# Proteomic identification of the UDP-GlcNAc: PI α1–6 GlcNAc-transferase subunits of the glycosylphosphatidylinositol biosynthetic pathway of *Trypanosoma brucei*

**Zhe Ji, Michele Tinti, Michael A. J. Ferguson** *

The Wellcome Centre for Anti-Infectives Research, School of Life Sciences, University of Dundee, Dundee, United Kingdom

* majferguson@dundee.ac.uk

## Abstract

The first step of glycosylphosphatidylinositol (GPI) anchor biosynthesis in all eukaryotes is the addition of N-acetylglucosamine (GlcNAc) to phosphatidylinositol (PI) which is catalysed by a UDP-GlcNAc: PI α1–6 GlcNAc-transferase, also known as GPI GnT. This enzyme has been shown to be a complex of seven subunits in mammalian cells and a similar complex of six homologous subunits has been postulated in yeast. Homologs of these mammalian and yeast subunits were identified in the *Trypanosoma brucei* predicted protein database. The putative catalytic subunit of the *T. brucei* complex, TbGPI3, was epitope tagged with three consecutive c-Myc sequences at its C-terminus. Immunoprecipitation of TbGPI3-3Myc followed by native polyacrylamide gel electrophoresis and anti-Myc Western blot showed that it is present in a ~240 kDa complex. Label-free quantitative proteomics were performed to compare anti-Myc pull-downs from lysates of TbGPI3-3Myc expressing and wild type cell lines. TbGPI3-3Myc was the most highly enriched protein in the TbGPI3-3Myc lysate pull-down and the expected partner proteins TbGPI15, TbGPI19, TbGPI2, TbGPI1 and TbERI1 were also identified with significant enrichment. Our proteomics data also suggest that an Arv1-like protein (TbArv1) is a subunit of the *T. brucei* complex. Yeast and mammalian Arv1 have been previously implicated in GPI biosynthesis, but here we present the first experimental evidence for physical association of Arv1 with GPI biosynthetic machinery. A putative E2-ligase has also been tentatively identified as part of the *T. brucei* UDP-GlcNAc: PI α1–6 GlcNAc-transferase complex.

## Introduction

*Trypanosoma brucei* is a protozoan pathogen that undergoes a complex life cycle between its tsetse fly vector and mammalian hosts. The parasite causes human African trypanosomiasis in humans and nagana in cattle in sub-Saharan Africa.

The bloodstream form (BSF) of *T. brucei* produces a dense coat of GPI anchored variant surface protein (VSG) to protect it from the innate immune system and, through antigenic variation, the acquired immune response [1]. Other *T. brucei* surface molecules that have been

**Data Availability Statement:** The data underlying this study are available at the PRIDE database through accession number PXD022979.

**Funding:** Z.J. China Scholarship Council PhD scholarship (201706310166), https://www.csc.edu.cn/ M.A.J.F. Wellcome Investigator Award (101842/Z13/Z), https://wellcome.org/.

**Competing interests:** The authors have declared that no competing interests exist.

shown experimentally to possess a GPI membrane anchor are the ESAG6-subunit of the BSF transferrin receptor (TfR) [2] and the procyclins, the major surface glycoproteins of the tsetse mid-gut dwelling procyclic form (PCF) of the parasite [3]. In addition, many other surface molecules with N-terminal signal peptides and C-terminal GPI addition signal peptides are predicted to be GPI-anchored in *T. brucei*, including the BSF haptaglobin-haemaglobin receptor [4] and the factor H receptor [5], the epimastigote BARP glycoprotein [6] and the metacyclic trypomastigote invariant surface protein (MISP) [7]. Thus far, GPI anchor structures have been completely or partially solved for four *T. brucei* VSGs [8–11], the TfR [2] and the procyclins [3]. As for the structure of GPIs, research on *T. brucei* was the first to yield methodologies to delineate the steps of GPI biosynthesis that were subsequently applied to mammalian cells and yeast [12–14]. However, it was the power of mammalian cell and yeast genetics that led to the identification of the majority of GPI biosynthesis genes, reviewed in [15–17].

We currently have reasonably advanced models for GPI anchor biosynthesis and processing in trypanosomes, mammalian cells and yeast and the similarities and differences in these pathways have been reviewed extensively elsewhere [15–18]. For most organisms, the functions and interactions of putative GPI pathway gene products have been inferred from experimental work in mammalian or yeast cells. In a few cases these functions have been experimentally confirmed in *T. brucei*, i.e., for the GlcNAc-PI de-N-acetylase (TbGPI12) [19], the third mannosyltransferase (TbGPI10) [20] and the catalytic (TbGPI8) [21] and other subunits (TTA1 and 2 [22] and TbGPI16 [23]) of the GPI transamidase complex.

The first step of GPI biosynthesis is the addition of GlcNAc to PI by a UDP-GlcNAc: PIα1–6 GlcNAc-transferase complex. The composition of this complex was determined in mammalian cells, where seven subunits have been identified: PIGA, PIGC, PIGH, PIGP, PIGQ, PIGY, a homologue of ERI1 first identified in yeast and shown to associate with GPI2 [24], and DPM2 (Table 1) [15]. The DPM2 component is a non-catalytic subunit of dolichol phosphate mannose synthetase. The complex was realised through a series of elegant functional cloning and co-immunoprecipitation experiments using individually epitope-tagged bait and prey components. A similar multi-subunit complex has been proposed in yeast where homologues for all the subunits, except DPM2, have been identified (Table 1) [17]. However, experimental evidence for physical associations between most of these yeast subunits is lacking.

Here we describe epitope tagging of the putative catalytic subunit of *T. brucei* UDP-Glc-NAc: PIα1–6 GlcNAc-transferase (TbGPI3), equivalent to yeast GPI3 and mammalian PIGA.

**Table 1. Genes encoding UDP-GlcNAc: PI α1–6 GlcNAc-transferase (GPI GnT) complex subunits in mammalian cells, yeast and *T. brucei*.**

| Mammalian gene | Yeast gene | *T. brucei* gene (TriTrypDB gene ID) | Calculated MW of *T. brucei* gene product | Rank in TbGPI3-3Myc pull-downs[1] | Rank in total *T. brucei* cell Proteome[2] |
|---|---|---|---|---|---|
| *PIGA* | *GPI3* | *TbGPI3* (Tb927.2.1780) | 51.6 kDa | 1 | 6475 |
| *PIGH* | *GPI15* | *TbGPI15* (Tb927.5.3680) | 28.8 kDa | 2 | 7038 |
| *PIGP* | *GPI19* | *TbGPI19* (Tb927.10.10110) | 16.5 kDa | 3 | 6185 |
| *PIGC* | *GPI2* | *TbGPI2* (Tb927.10.6140) | 38.5 kDa | 4 | 6798 |
| *PIGQ* | *GPI1* | *TbGPI1* (Tb927.3.4570) | 81.5 kDa | 5 | 5285 |
| | | *TbArv1* (Tb927.3.2480) | 28.7 kDa | 6 | 7149 |
| *PIGY* | *ERI1* | *TbERI1* (Tb927.4.780) | 10.1 kDa | 7 | 7149 |
| | | *TbUbCE* (Tb927.2.2460) | 22.6 kDa | 8 | 2538 |
| *DPM2* | - | *TbDPM2* (Tb927.9.6440)[3] | 9.3 kDa | not present | 7149 |

[1]Taken from (S2 File).

[2]Taken from (S1 File) where proteins are ranked from 1 (most abundant) to 7148 (least abundant) and undetected proteins are ranked 7149.

[3]Probable mis-annotation, see text.

We demonstrate its presence in a protein complex and identify its partner proteins through label-free quantitative proteomics.

## Materials and methods

### Cultivation of trypanosomes

*T. brucei brucei* Lister strain 427 bloodstream form (BSF) parasites expressing VSG variant 221 and transformed to stably express T7 polymerase and the tetracycline repressor protein under G418 antibiotic selection [25] was used in this study and will be referred as bloodstream form wild type (BSF WT). Cells were cultivated in HMI-11T medium containing 2.5 µg/mL of G418 at 37°C in a 5% $CO_2$ incubator as previously described [25]. HMI-11T is a modification of the original HMI-9 [26] that uses 56 mM 1-thioglycerol in place of 200 mM 2-mercaptoethanol, and contains 10% heat inactivated fetal bovine serum (PAA) and lacks of serum plus (Hazleton Biologics, Lenexa, Kansas).

### DNA isolation and manipulation

Plasmid DNA was purified from *Escherichia coli* (chemically competent DH5α cells) using Qiagen Miniprep kits and Maxiprep was performed by the University of Dundee DNA sequencing service. Gel extraction and PCR purification were performed using QIAquick kits (Qiagen). Custom oligonucleotides were obtained from Eurofins MWG Operon or Thermo Fisher. *T. brucei* genomic DNA was isolated from ~$5 \times 10^7$ BSF cells using lysis buffer containing 100 mM Tris-HCl (pH 8.0), 100 mM NaCl, 25 mM EDTA, 0.5% SDS, and 0.1 mg/mL proteinase K (Sigma) by standard methods.

### Generation of in-situ tagging constructs

The tagging cassette was amplified from the pMOTag43M plasmid [27] using the forward primer: 5′–TGATTGATATTGCACCAGATTTTCCACTGGAGTTGTACTCTCGTAACCGGGA GAAGCTTCAAGTTGTGGGAAGCCCATCCgaacaaaagctgggtacc-3′ and the reverse primer: 5′–CAACGCGAAACAATGACAgAGAGAGAGAGAGAAGGGCGAAAACAAAAAGGAT CGCGGTAGAGAGGACCCCGCCCATACCCctattcctttgccctcggac-3′. The PCR product contains 80 bp corresponding to the 3'-end of the TbGPI3 open reading frame (capital letters of forward primer) followed by a sequence encoding the 3Myc epitope tag, an intergenic region (igr) from the *T. brucei* α-β tubulin locus, the hygromycin phosphotransferase (HYG) selectable marker gene and the 3'-UTR of TbGPI3 (capital letters of reverse primer).

### Transformation of BSF *T. brucei*

Constructs for *in situ* tagging were purified and precipitated, washed with 70% ethanol, and re-dissolved in sterile water. The released DNA was electroporated into BSF WT cell line. Cell culture and transformation were carried out as described previously [25, 27]. After five days of selection with hygromycin, cells were sub-cloned and four independent clones were selected and cultured.

### Western blot of cell lysates

To confirm the C-terminal tagging of TbGPI3 with 3Myc, cells from the four selected clones in parallel with BSF WT cells were lysed in SDS sample buffer. Aliquots corresponding to $5 \times 10^6$ cells per sample, were subjected to SDS-PAGE on NuPAGE bis-Tris 10% acrylamide gels (Invitrogen) and transferred to a nitrocellulose membrane (Invitrogen). Ponceau staining confirmed equal loading and transfer. The blot was further probed with anti-Myc rat monoclonal

antibody (Chromotek, 9E1) in a 1:1,000 dilution. Detection was carried out using IRDye 800CW conjugated goat anti-rat IgG antibody (1:15,000) and LI-COR Odyssey infrared imaging system (LICOR Biosciences, Lincoln, NE).

## Co-immunoprecipitation and Native-PAGE Western blotting

To investigate detergent solubilisation conditions for the immunoprecipitation of TbGPI3-3Myc complexes, aliquots of $2 \times 10^8$ cells were harvested and lysed in 500 μL of 50 mM Tris-HCl, pH 7.4, 150 mM NaCl containing different detergents; 0.5% digitonin, 1% digitonin, 1% Triton X-100 (TX-100), 1% n-octyl-beta-glucoside (NOG) or 1% decyl-β-D-maltopyranoside (DM). After centrifugation at 16,000 g, 4˚C for 20 min, aliquots of the supernatants equivalent to 2 x $10^8$ cells were incubated with 10 μL anti-Myc agarose beads (Myc-Trap™, Chromotek) for 1 h at 4˚C. The beads were washed three times in 50 mM Tris-HCl, pH 7.4, 150 mM NaCl containing the corresponding detergents and bound proteins were eluted three times with 10 μL 0.5 mg/mL c-Myc peptide (Sigma M2435) in the corresponding detergent containing buffer. The combining eluates for each detergent condition, equivalent to $2 \times 10^8$ cells, were subjected to NativePAGE (Invitrogen) and transferred to a PVDF membrane (Invitrogen) followed by immunoblotting with anti-Myc antibody (Chromotek, 9E1) diluted 1:1,000. The blot was then developed by ECL using an HRP-conjugated secondary antibody (Sigma, A9037, 1:3,000).

## Label free proteomics of TbGPI3-3Myc and BSF WT lysate pull downs

BSF WT and TbGPI3-3Myc expressing cell lines were cultured and $1 \times 10^9$ cells of each were harvested and lysed in 1 mL of lysis buffer containing 0.5% digitonin. After centrifugation 16,000 g, 4˚C for 20 min, the supernatants were mixed with 20 μL of Myc-Trap™ beads and incubated for 1 h at 4˚C. The beads were washed three times in the same buffer, and bound proteins were eluted with 1×SDS sample buffer and subjected to SDS-PAGE, running the proteins only 10 cm into the gel. Whole lanes containing TbGPI3 and wild type cell lines samples were cut identically into 3 slices and the gel pieces were dried in Speed-vac (Thermo Scientific) for in-gel reduction with 0.01 M dithiothreitol and alkylation with 0.05 M iodoacetamide (Sigma) for 30 min in the dark. The gel slices were washed in 0.1 M $NH_4HCO_3$, and digested with 12.5 μg/mL modified sequence grade trypsin (Roche) in 0.02 M $NH_4HCO_3$ for 16 h at 30˚C. Samples were dried and re-suspended in 50 μL 1% formic acid and then subjected to liquid chromatography on Ultimate 3000 RSLC nano-system (Thermo Scientific) fitted with a 3 Acclaim PepMap 100 (C18, 100 μM × 2 cm) and then separated on an Easy-Spray PepMap RSLC C18 column (75 μM × 50 cm) (Thermo Scientific). Samples (15μL) were loaded in 0.1% formic acid (buffer A) and separated using a binary gradient consisting of buffer A and buffer B (80% acetonitrile, 0.1% formic acid). Peptides were eluted with a linear gradient from 2 to 35% buffer B over 70 min. The HPLC system was coupled to a Q-Exactive Plus Mass Spectrometer (Thermo Scientific) equipped with an Easy-Spray source with temperature set at 50˚C and a source voltage of 2.0 kV. The mass spectrometry proteomics data have been deposited to the ProteomeXchange Consortium via the PRIDE partner repository with the dataset identifier Project PXD022979 [28] https://www.ebi.ac.uk/pride/archive/projects/PXD022979.

## Protein identification by MaxQuant

RAW data files were analysed using MaxQuant version 1.6.10.43, with the in-built Andromeda search engine [29], using the *T. brucei brucei* 927 annotated protein sequences from Tri-TrypDB release 46 [30], supplemented with the *T. brucei brucei* 427 VSG221 (Tb427. BES40.22) protein sequence. The mass tolerance was set to 4.5 ppm for precursor ions and

MS/MS mass tolerance was set at 20 ppm (MaxQuant default parameters). The enzyme was set to trypsin, allowing up to 2 missed cleavages. Carbamidomethyl on cysteine was set as a fixed modification. Acetylation of protein N-termini, and oxidation of methionine were set as variable modifications. Match between runs was enabled, allowing transfer of peptide identifications of sequenced peptides from one LC-MS run to non-sequenced ions with the same mass and retention time in another run. A 20-min time window was set for alignment of separate LC-MS runs. The false-discovery rate for protein and peptide level identifications was set at 1%, using a target-decoy based strategy.

### Data analysis

Data analysis was performed using custom Python scripts, using the SciPy ecosystem of open-source software libraries [31]. The data analysis pipeline is available at GitHub https://github.com/mtinti/TbGPI3 and Zenodo https://zenodo.org/record/4310034 repositories, DOI:10.5281/zenodo.3735036. The MaxQuant proteinGroups.txt output file was used to extract the iBAQ scores for forward trypanosome protein sequences identified with at least two unique peptides and with an Andromeda score >4. The protein iBAQ scores were normalised for sample loading by dividing each iBAQ value by the median of all the iBAQ values in each experiment. Missing values were replaced by the smallest iBAQ value in each sample. Differential abundance analysis between the bait and control samples was performed with the ProtRank Python package [32]. Briefly, ProtRank performs a rank test between each control and bait sample pair to output as signed-rank and false discovery rate values. The signed-rank is proportional to the significance of the differential abundance of the protein groups between the bait and control samples.

The BSF protein intensity (abundance) rank (S1 File) was computed from a recent dataset published by our laboratory [33] of *T. brucei* protein half-lives computed from a label-chase experiment. In those experiments, BSF parasites were labelled to steady-state in medium SILAC culture medium (M) and then placed into light SILAC culture medium (L). Seven time points, with three biological replicates, were sampled and each mixed 1:1 with BSF lysate labelled to steady state in heavy SILAC culture medium (H) to provide an internal standard for normalisation. Each sample was then separated into 10 sub-fractions for LC-MS/MS (thus, a total of 210 LC-MS/MS analyses were performed). Here, we exploited the heavy-labelled internal standard in every sample: The log10 summed eXtracted Ion Currents (XICs) of the heavy-labelled peptides for each protein were averaged across the BSF replicates and used to rank a very deep BSF proteome from the most abundant (rank = 1) to the least abundant (rank = 7148). Undetected proteins were given a rank of 7149.

## Results

### Identification of putative *T. brucei* UDP-GlcNAc: PI α1–6 GlcNAc-transferase complex components

Conventional BLASTp searches with default settings [34] were sufficient to identify *T. brucei* homologues of PIGA(GPI3), PIGC(GPI2), PIGP(GPI19), PIGQ(GPI1) and DPM2. However, the results for PIGH(GPI15) and PIGY(ERI1) were equivocal so a Domain Enhanced Lookup Time Accelerated BLAST [35] using a PAM250 matrix was applied to find the corresponding *T. brucei* homologues (Table 1).

### *In situ* epitope tagging of TbGPI3

To investigate whether a multi-subunit UDP-GlcNAc: PI α1–6 GlcNAc-transferase complex might exist in *T. brucei* we selected TbGPI3, which encodes a 455 amino acid protein with two

predicted transmembrane domains, one near its N-terminus and one near its C-terminus [36], for epitope tagging. We chose the PIGA(GPI3) homologue as the bait protein because PIGA has been shown to have either direct or indirect interactions with all other subunits in the mammalian UDP-GlcNAc: PI α1–6 GlcNAc-transferase complex [15]. Alignment of putative TbGPI3, yeast GPI3 and PIGA protein sequences show that the *T. brucei* sequence has 43.9% and 50.8% sequence identity with the yeast and human sequences, respectively (S1 Fig).

*In situ* tagging of the TbGPI3 gene was achieved by transfecting BSF *T. brucei* with PCR products amplified from the pMOTag43M plasmid [27] (Fig 1A). Transfected cells were selected using hygromycin and subsequently cloned by limit-dilution. Lysates of four separate clones were subjected to anti-Myc Western blotting (Fig 1B and 1C). *In situ* tagged TbGPI3-3Myc protein was detected in all four clones at an apparent molecular weight of ~47 kDa, somewhat lower than the predicted molecular weight of 55 kDa (Fig 1C, lanes 1–4). The specificity of this staining, although weak, is illustrated by the absence of comparable staining for BSF WT sample (Fig 1C, lane 5). The weakness of the TbGPI3-3Myc staining with anti-Myc is a function of the extremely low abundance of TbGPI3 in the total cell proteome, where it ranks $6475^{th}$ out of 7148 detectable protein groups (Table 1), and the limitations of protein loading for BSF cell lysates on SDS-PAGE caused by the abundance of VSG and tubulin in these cells (S1 File) from their surface coat and subpellicular cytoskeleton, respectively. These (glyco)proteins can be seen running above and below the 50 kDa marker in (Fig 1B).

## Solubilisation and native-PAGE of TbGPI3-3Myc

The analysis of epitope-tagged membrane bound multiprotein complexes requires detergent extraction and anti-epitope pull-down under conditions that preserve intermolecular interactions within the complex. To investigate detergent extraction conditions, TbGPI-3Myc expressing cells were cultured and lysed with 0.5% digitonin, 1% digitonin, 1% TX-100, 1% NOG and 1% DM and centrifuged. The solubilised proteins in the supernatants from these treatments, along with a 1% TX-100 extract of wild type cells, were immunoprecipitated with Myc-Trap$^{TM}$ agarose beads that were washed three times and finally eluted with synthetic c-Myc peptide. The proteins in the eluates were separated by denaturing SDS-PAGE and by native PAGE [37] and analysed by anti-Myc Western blot (Fig 2A and 2B, respectively). All detergents, apart from DM, extracted the TbGPI3-3Myc protein (Fig 2A). Of these conditions, 1% TX-100 gave the highest efficiency of extraction but duplicate samples analysed by native PAGE and anti-Myc Western blot showed that digitonin best preserved a TbGPI3-3Myc-containing complex with a native apparent molecular weight of ~240 kDa and that 0.5% digitonin gave a higher-yield of the complex than 1% digitonin (Fig 2B). The reason for not detecting any clear complexes in other conditions may be due to the ~240 kDa complex falling apart into multiple sub-complexes below the limits of detection. In all of these experiments, we adopted a pull-down (from $2 \times 10^8$ cell equivalents) prior to immunodetection approach because of the aforementioned low abundance of TbGPI3 and its expected partner proteins (Table 1) and the correspondingly weak anti-Myc signals obtainable from $5 \times 10^6$ cell equivalents (Fig 1C).

## Identification of *T. brucei* UDP-GlcNAc: PI α1–6 GlcNAc-transferase complex components by quantitative proteomics

Having established detergent solubilisation conditions that retained TbGPI13-3Myc in a complex, we performed label-free quantitative proteomics on Myc-Trap$^{TM}$ pull downs to identify the components within the complex. For this experiment, BSF WT and TbGPI13-3Myc expressing parasites were grown under identical conditions and the same numbers of cells

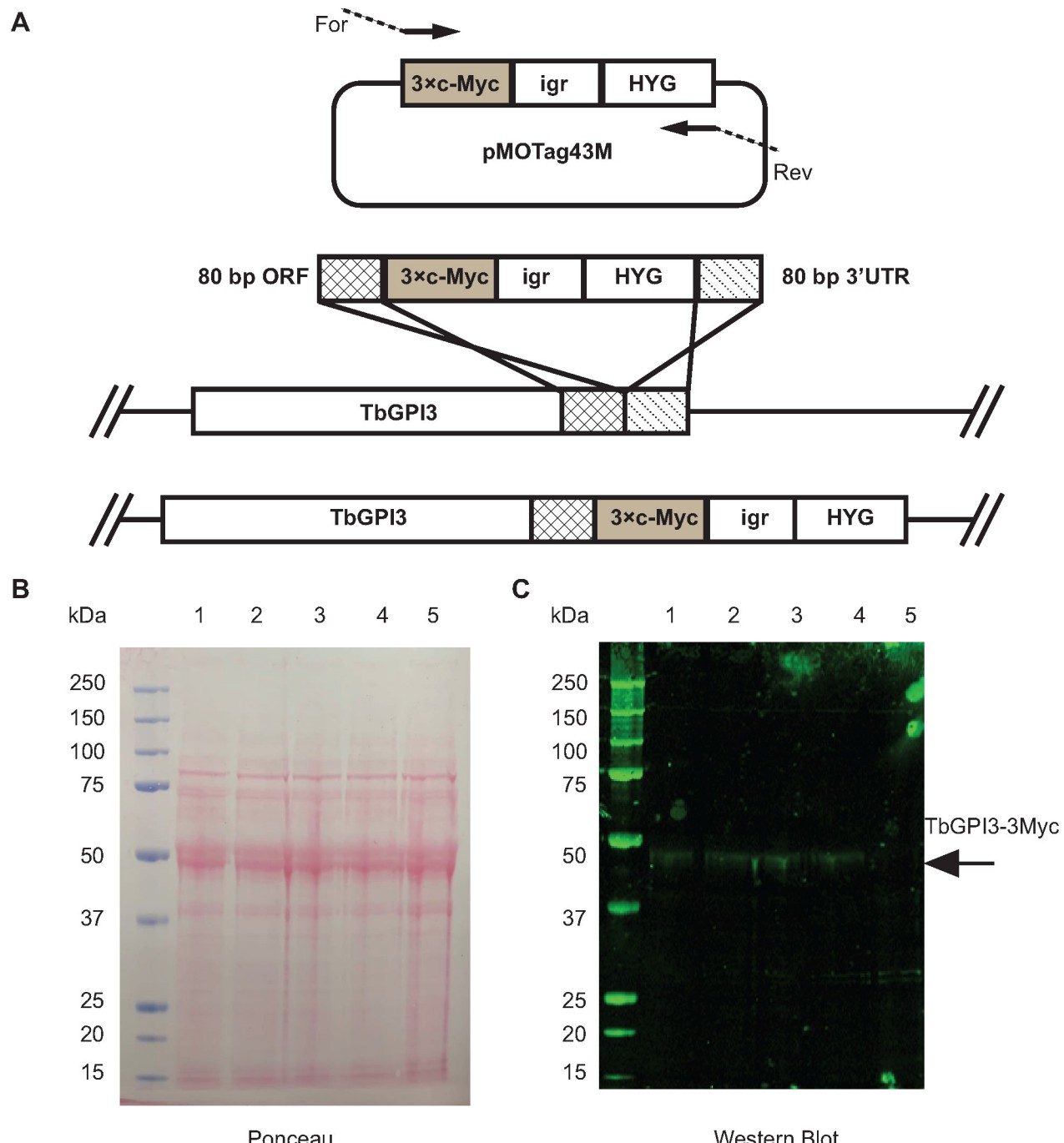

**Fig 1. *In situ* C-terminal tagging of TbGPI3 with 3Myc.** (A) Map of plasmid pMOTag43M [27] used for the *in situ* tagging of TbGPI3, and a scheme of how the PCR product generated with the indicated forward (For) and reverse (Rev) primers inserts into the 3'-end of the *TbGPI3* ORF (checked box) and 3'-UTR (striped box) in the parasite genome to effect *in-situ* tagging. HYG = hygromycin phosphotransferase selectable marker; igr = α-β tublin intergenic region. (B) Ponceau staining of denaturing SDS-PAGE Western blot shows similar loading and transfer of lysates (corresponding to $5 \times 10^6$ cells) from four *in-situ* tagged clones (lanes 1–4) and wild type cells (lane 5). (C) The identical blot was probed with anti-Myc antibody. TbGPI3-3Myc is indicated by the arrow. The positions of molecular weight markers are indicated on the left of (B) and (C).

were harvested and lysed in 0.5% digitonin lysis buffer. Immunoprecipitation was performed using Myc-Trap™ beads and the proteins eluted from these two samples with SDS sample buffer were processed to tryptic peptides for LC-MS/MS analysis (Fig 3A). The experiments

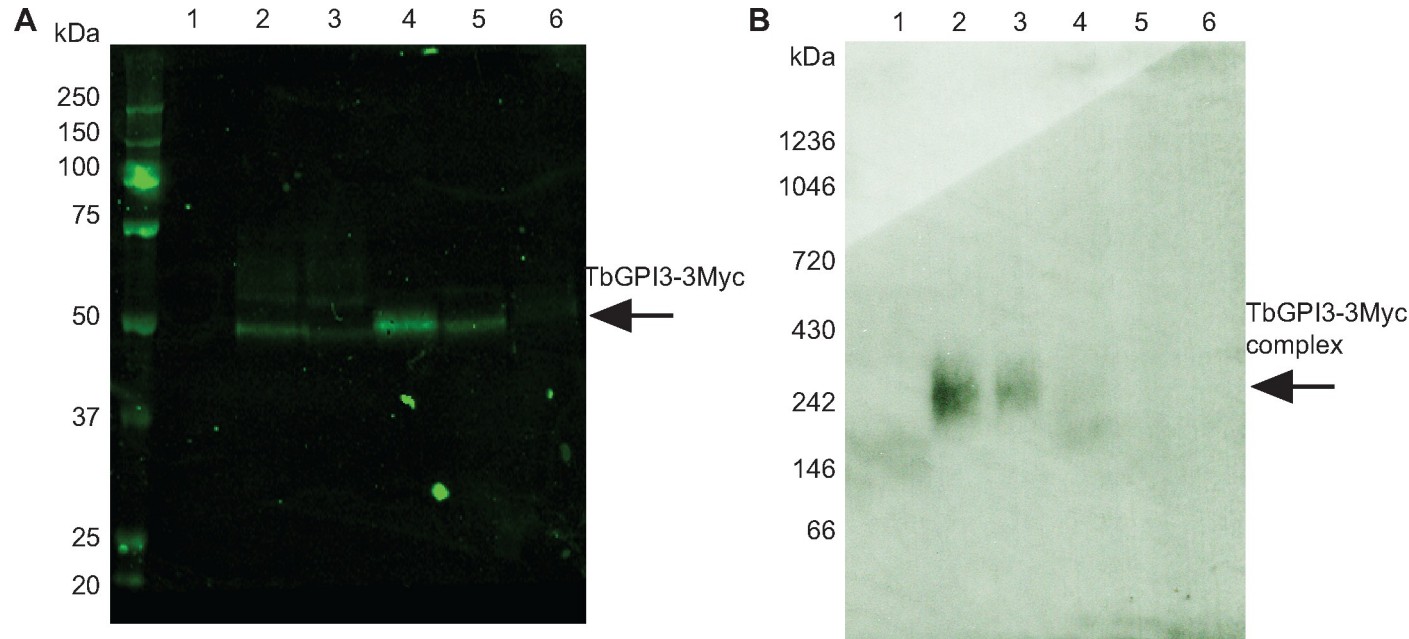

**Fig 2. TbGPI3-3Myc is present in complexes in BSF *T. brucei*.** (A) Aliquots of $2 \times 10^8$ cells were harvested and lysed in lysis buffer containing different detergents to assess TbGPI3-3Myc solubilisation. After immunoprecipitation of the supernatants with anti-Myc agarose beads, proteins were eluted with 0.5 mg/mL c-Myc peptide and aliquots were subjected to SDS-PAGE followed by anti-Myc Western blotting. (B) Identical samples were also separated by native-PAGE and subjected to anti-Myc Western blotting. In both cases, lane 1 corresponds to wild type cells lysed with 1% TX-100, as a negative control for anti-Myc blotting, and lanes 2–6 correspond to TbGPI3-3Myc clone1 lysed with 0.5% digitonin, 1% digitonin, 1% TX-100, 1% NOG or 1% DM, respectively.

were performed in biological triplicates and the data were analysed using MaxQuant software and a newly developed data analysis method written in Python called ProtRank [32], see Materials and Methods. The protein groups identified (S2 File) were displayed on a plot of the minus log10 value of their False Discovery Rate (y-axis) and enrichment rank (x-axis) between the bait versus control samples (Fig 3B). As expected, the bait protein TbGPI3-3Myc was the most highly enriched protein and its putative partner proteins TbGPI15, TbGPI19, TbGPI2, TbGPI1 and TbERI1 were also significantly enriched and present in the top-7 proteins in the pull-downs (Table 1). Notably, TbDPM2 (dolichol-phosphate-mannose synthetase 2) was not detected. However, although TbDPM2 is annotated in the TriTrypDB database it should be noted that, like yeast, *T. brucei* makes a single-chain dolichol-phospho-mannose synthetase (DPM1) [38] rather than a trimeric enzyme made of a soluble catalytic DPM1 subunit associated with small transmembrane DPM2 and DPM3 subunits, as found in mammalian cells. For these reasons, we feel that the absence of DPM protein components in the *T. brucei* complex is to be expected.

Interestingly, an Arv1-like protein (hereon referred to as TbArv1, Tb927.3.2480) and a putative ubiquitin-conjugating enzyme E2 (UbCE, Tb927.2.2460) were also co-immunoprecipitated with TbGPI3 (Fig 3B). The data were also processed in a different way (see Materials and Methods) that plots the experimental rank (x-axis) against the rank order of estimated abundances of the protein groups in the total cell proteome (S1 File), generated from data in [33], on the y-axis (Fig 3C). In this plot, the very low abundance TbArv1 (undetectable in the total cell proteome) clusters well with the canonical and similarly low abundance UDP-Glc-NAc: PI α1–6 GlcNAc-transferase subunits. By contrast, although UbCE is clearly highly-enriched in the pull-down it is a much more abundant protein, suggesting that only some fraction of it may be associated with the complex.

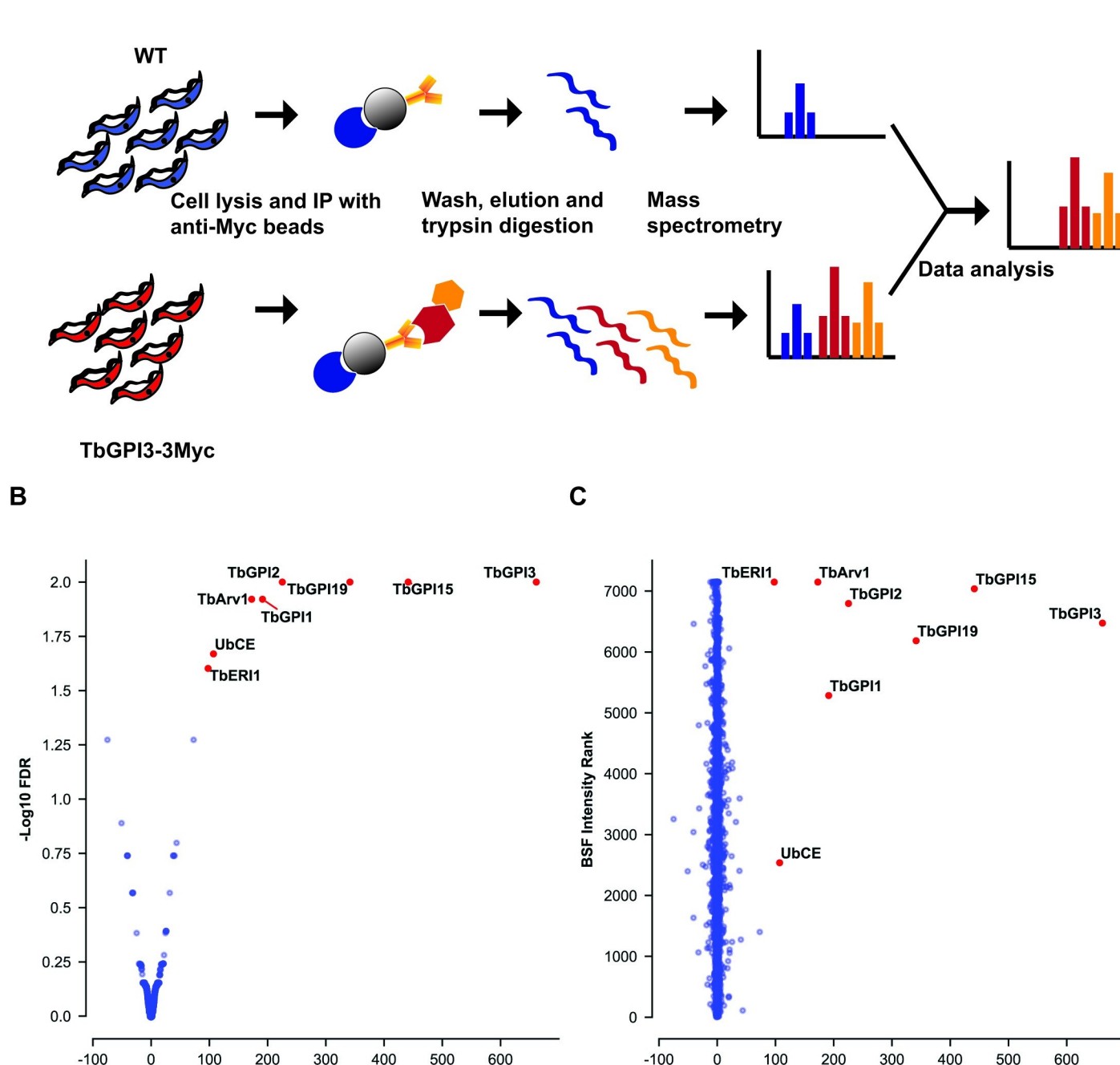

**Fig 3. Identification of UDP-GlcNAc: PI α1–6 GlcNAc-transferase subunits by immunoprecipitation of TbGPI3-3Myc from BSF *T. brucei* digitonin lysates.** (A) Scheme of the label-free proteomics approach to identify TbGPI13-3Myc binding partners. BSF WT and TbGPI13-3Myc expressing cell lines were cultured, harvested and lysed in 0.5% digitonin lysis buffer. Identical quantities of the supernatants were subjected to anti-Myc agarose bead immunoprecipitation and the bound proteins were eluted from the beads with SDS sample buffer. The eluted proteins were reduced, alkylated and digested with trypsin and the resulting peptides analysed by LC-MS/MS. (B) Volcano plot comparing protein groups present in the anti-cMyc immunoprecipitates from TbGPI3-3Myc expressing cell lysates versus WT cell lysates. Mean values (from biological triplicate experiments) for each protein group (dots) are plotted according to their minus log10 False Discovery Rate values (y-axis), calculated by MaxQuant, and the enrichment rank (x-axis). The enrichment rank was computed with the ProtRank algorithm using the iBAQ values calculated by MaxQuant. The higher the rank value on the x-axis, the higher the abundance in the TbGPI3-3Myc samples. The putative subunits of UDP-GlcNAc: PI α1–6 GlcNAc-transferase in *T. brucei* are highlighted in red and annotated with their corresponding names (Table 1). (C) Relative intensity plot using a new algorithm. The same data as (B) are plotted with a different y-axis, whereby each protein group is assigned an intensity rank from the most abundant protein group (1) to least

abundant protein groups (7148) based on their summed eXtracted Ion Currents (XICs) for the total BSF proteome. (Details of the mass spectrometry and data analysis are provided in in Materials and Methods).

## Discussion

The proteomics data herein suggest that: (i) The *T. brucei* UDP-GlcNAc: PI α1–6 GlcNAc-transferase complex (GPI GnT) contains the expected subunits TbGPI3, TbGPI15, TbGPI19, TbGPI2, TbGPI1 and TbERI1. (ii) Like yeast, but unlike mammalian cells, DPM components are not subunits of the parasite complex. (iii) An Arv1-like protein (TbArv1) is a part of the parasite complex. (iv) A putative E2-ligase UbCE may be a part of the parasite complex.

The limitations of this study are that it does not attempt to assess stoichiometry or topology of the *T. brucei* UDP-GlcNAc: PI α1–6 GlcNAc-transferase complex components and that it lacks an orthogonal confirmation of the presence of TbArv1 and TbUbCE in the complex. With respect to the latter we rely, instead, on the extremely high and reproducible enrichment of these two components in the TbGPI3-3Myc pull-downs: From undetectable (>7148[th]) and 2538[th] in the total cell proteome to 6[th] and 7[th] in the TbGPI3-3Myc pull-downs, respectively.

TbArv1 is predicted to contain four transmembrane domains and an Arv1 domain (PF04161). Previous studies in yeast have indicated that Arv1p, although non-essential for growth and therefore GPI-anchoring of proteins at 25˚C [39, 40], is required for the efficient synthesis of $Man_1GlcN$-acylPI (mannosyl-glucosaminyl-acyl-phosphatidylinositol) [41] It has been postulated to be a GPI flippase [41, 42] helping deliver GlcN-acylPI, which is made on the cytoplasmic face of the ER, to the active site of mannosyl-transferase I (MT I) on the luminal face of the ER. The complementation of yeast Arv1 mutants by the human Arv1 [43] and recent findings that human Arv1 mutations lead to deficiencies in GPI anchoring [44, 45] strongly suggest a related role in mammalian cells. However, whether it is a component of the mammalian and yeast UDP-GlcNAc: PI α1–6 GlcNAc-transferase complexes, or indeed of possible GlcNAc-PI de-N-acetylase of GPI flippase complexes, is unclear. It is possible that TbArv1 plays an analogous role to that proposed for Arv1 in yeast and mammalian cells in the *T. brucei* GPI pathway. However, since (unlike yeast and mammalian cells) acylation of the PI moiety occurs strictly after the action of MT I in *T. brucei* [46], TbArv1 would need to facilitate the delivery of GlcN-PI rather than GlcN-acylPI to TbMT I in this parasite. Further, it is worth noting that unlike mammalian cells and yeast, which are thought to only translocate GlcN-acylPI, *T. brucei* appears to flip-flop most of its GPI intermediates between the cytoplasmic and lumenal surfaces of the ER [47]. Thus, it is possible that *T. brucei* Arv1 protein, given its location, may play some other, perhaps regulatory, role in the UDP-GlcNAc: PI α1–6 GlcNAc-transferase reaction. This may also be the case in mammalian and yeast cells.

Finally, a recent study in mammalian cells showed that GPI anchor biosynthesis is upregulated in ERAD (endoplasmic-reticulum-associated protein degradation) deficient and PIGS mutant cell lines, suggesting that the GPI anchor biosynthetic pathway is somehow linked to and regulated by the ERAD system [48]. Since ERAD involves E2-dependent ubiquitylation of misfolded proteins as they exit the ER, it is possible that the UbCE associated, in part, with UDP-GlcNAc: PI α1–6 GlcNAc-transferase complex might play some role in regulation of the *T. brucei* GPI pathway.

## Supporting information

**S1 Fig. Sequence alignment of GPI3, PIGA and TbGPI3.**
(PDF)

**S1 Graphical abstract.**
(TIF)

**S1 File. Total cell proteome of bloodstream form *T. brucei* ranked by intensity.**
(XLSX)

**S2 File. Label free quantitative proteomics data.**
(XLSX)

**S1 Raw images. Raw images of Figs 1B, 1C, 2A and 2B.**
(PDF)

## Acknowledgments

We thank Drs Alvaro Acosta Serrano, Lucia Guther and Samuel Duncan for helpful advice and the Fingerprints Proteomics Facility for expert assistance with quantitative proteomics.

## Author Contributions

**Conceptualization:** Zhe Ji, Michele Tinti, Michael A. J. Ferguson.

**Data curation:** Zhe Ji, Michele Tinti, Michael A. J. Ferguson.

**Formal analysis:** Zhe Ji, Michele Tinti, Michael A. J. Ferguson.

**Funding acquisition:** Michael A. J. Ferguson.

**Investigation:** Zhe Ji, Michele Tinti, Michael A. J. Ferguson.

**Methodology:** Zhe Ji, Michele Tinti, Michael A. J. Ferguson.

**Project administration:** Michael A. J. Ferguson.

**Resources:** Michael A. J. Ferguson.

**Supervision:** Michael A. J. Ferguson.

**Validation:** Zhe Ji, Michele Tinti.

**Visualization:** Zhe Ji, Michele Tinti, Michael A. J. Ferguson.

**Writing – original draft:** Zhe Ji, Michele Tinti, Michael A. J. Ferguson.

**Writing – review & editing:** Zhe Ji, Michele Tinti, Michael A. J. Ferguson.

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
