## [Decision Letter · Decision Letter 0]

5 Jan 2021

PONE-D-20-39015

Proteomic identification of the UDP-GlcNAc : PI α1-6 GlcNAc-transferase subunits of the glycosylphosphatidylinositol biosynthetic pathway of *Trypanosoma bruce*

PLOS ONE

Dear Dr. Ferguson,

Thank you for submitting your manuscript to PLOS ONE. After careful consideration, we feel that it has merit but does not fully meet PLOS ONE’s publication criteria as it currently stands. Therefore, we invite you to submit a revised version of the manuscript that addresses the points raised by the reviewers (see below).

We look forward to receiving your revised manuscript.

Kind regards,

Ziyin Li, Ph.D.

Academic Editor

PLOS ONE

Journal Requirements:

2)  Please provide the source of the T. brucei strain used in your study.

3) Please ensure you have discussed any potential limitations of your study in the Discussion.

4) PLOS ONE now requires that authors provide the original uncropped and unadjusted images underlying all blot or gel results reported in a submission’s figures or Supporting Information files. This policy and the journal’s other requirements for blot/gel reporting and figure preparation are described in detail at https://journals.plos.org/plosone/s/figures#loc-blot-and-gel-reporting-requirements and https://journals.plos.org/plosone/s/figures#loc-preparing-figures-from-image-files. When you submit your revised manuscript, please ensure that your figures adhere fully to these guidelines and provide the original underlying images for all blot or gel data reported in your submission. See the following link for instructions on providing the original image data: https://journals.plos.org/plosone/s/figures#loc-original-images-for-blots-and-gels.

5) Please amend either the title on the online submission form (via Edit Submission) or the title in the manuscript so that they are identical.

6) Please include your tables as part of your main manuscript and remove the individual files. Please note that supplementary tables (should remain/ be uploaded) as separate "supporting information" files

7) Please include captions for your Supporting Information files at the end of your manuscript, and update any in-text citations to match accordingly. Please see our Supporting Information guidelines for more information: http://journals.plos.org/plosone/s/supporting-information.

Reviewers' comments:

Reviewer's Responses to Questions

**Comments to the Author**

1. Is the manuscript technically sound, and do the data support the conclusions?

Reviewer #1: Yes

Reviewer #2: Yes

Reviewer #3: Yes

Reviewer #4: Yes

2. Has the statistical analysis been performed appropriately and rigorously? 

Reviewer #1: I Don't Know

Reviewer #2: Yes

Reviewer #3: Yes

Reviewer #4: Yes

3. Have the authors made all data underlying the findings in their manuscript fully available?

Reviewer #1: Yes

Reviewer #2: Yes

Reviewer #3: Yes

Reviewer #4: Yes

4. Is the manuscript presented in an intelligible fashion and written in standard English?

Reviewer #1: Yes

Reviewer #2: Yes

Reviewer #3: Yes

Reviewer #4: Yes

5. Review Comments to the Author

Reviewer #1: The manuscript is a bit thin and borders on the trivial. Surely the authors could have validated and extended their data to make the paper more generally useful to the field? Some suggestions:

(i) Arv1 is new here - an Arv1 knockdown strain could be generated to test whether the complex as seen on native page is affected in any way. This could also be done with one or other of the subunits - perhaps such strains are available and it would be a matter of expressing myc-tagged Gpi3 in these as a reporter. (ii) Is the complex stoichiometric - one copy of each subunit? The native page result (Fig 2) is suggestive, but not definitive. This could be investigated. (iii) PIG-A is a single-pass, tail anchored protein, but TbGpi3 appears to have two transmembrane sequences - this could be checked.

Minor comments:

line 78 - cite Sobering et al. (2004) Cell for additional subunit Eri1

Figure 1B, C - the Ponceau-stained membrane shows remarkably few bands for a total lysate, with the dominant band running exactly the same as the myc-tagged protein - this seems odd. Th blot in panel C is of quite poor quality - there are clearer results in Fig 2A. Figure 1 seems unnecessary as the key result in this figure is replicated in Fig 2A.

line 251 - it is not necessary to do a pull-down in order to detect complexes in native page - a total lysate could have been run, but perhaps the sample was too dilute?

paragraph beginning on line 334 - Arv1 is not essential in yeast cells grown at 25C (Swain, Stukey et al. (2002) JBC; Georgiev, Johansen et al. (2013) Traffic). Although Arv1 may play a role in GPI biosynthesis, perhaps a regulatory role, it is clearly not necessary as otherwise the cells would be inviable. This point should be made clear.

Reviewer #2: This manuscript reports results of a focused study on Trypanosoma brucei enzyme complex for the initial step in GPI biosynthesis. GPI-anchored proteins are most abundant proteins in the plasma membrane of T. brucei, a causative agent of human sleeping sickness and Nagana disease of the livestock. The corresponding human and yeast enzyme complexes were studied before and they consist of common components with some difference. Now authors report that the trypanosome’s enzyme contains all components that are common among human and yeast enzymes and that the trypanosome’s enzyme contained two new components, TbArv1 and UbCE. Yeast and human ARV1 were reported to be involved in GPI biosynthesis, however, its exact function has been unclear. The finding that TbArv1 is a component of the initial enzyme complex will facilitate clarification of its function. The association of an ERAD E2 enzyme UbCE with the initial enzyme of GPI biosynthesis pathway suggests the pathway might be under regulation by the ERAD. This manuscript, therefore, will provide a basis for further studies to better understand regulation of GPI biosynthesis in not only in T. brucei but also in human and yeast cells. The manuscript is clearly written and the conclusions are sufficiently supported by experimental evidence.

Reviewer #3: The manuscript written by Ji et al. described the initial step of glycosylphosphatidylinositol (GPI) biosynthetic pathway in Trypanosoma brucei. First, the authors bioinformatically compared the components of GPI N-acetylglucosamine transferase (GPI-GnT) among T. brucei, yeast, and mammals. Then, TbGPI3, which is the putative catalytic subunit of T. brucei GPI-GnT, was epitope-tagged, and immunoprecipitated to determine the partner proteins. The authors identified Arv1 and UbCE proteins as part of GPI-GnT, in addition to TbGPI15, TbGPI9, TbGPI2, TbGPI1, and TbERI1. It is a new finding that two additional factors are componets of GPI-GnT in T. brucei. However, the reviewer asks the authors to confirm the mass spectrometric result before publication.

The major concern is whether Arv1 and UbCE are bound with GPI-GnT directly. The authors only detected two proteins with mass spectrometry. The reviewer suggests to validate the mass spectrometric results. It is possible to detect the interactions between tagged Arv1 or UbCE and TbGPI3-3Myc using western blotting.

In lines 340 to 342, the authors wrote that “The complementation of yeast Arv1 mutants by the human Arv1 [38] and recent findings that human Arv1 mutations lead to deficiencies in GPI anchoring [39] [40] strongly suggest a related role in mammalian cells and that it is a component of the mammalian UDP-GlcNAc : PI 1-6 GlcNAc-transferase complex.” It is obvious that Arv1 is involved in the GPI biosynthetic pathway, but cannot specify to be the component of GPI-GnT from the reports. The authors should rewrite the part.

In yeast, it is reported that Arv1 is required for flipping of GPI intermediates or for efficient synthesis of Man1GlcN-acylPI (Okai et al. (2020) FEBS Lett. 594: 2431; Kajiwara et al. (2008) Mol. Biol. Cell 19: 2069). The difference in GPI biosynthetic pathway between yeast and T. brucei is the flipping steps of GPI intermediates. Only the GlcN-PI across the ER lipid bilayer in yeast GPI biosynthesis, whereas several GPI intermediates seem to be flipped/flopped in T. brucei. The authors need to describe the difference of the flipping reaction in yeast and T. brucei GPI biosynthesis, in addition to the difference of inositol-acylation and mannosylation steps.

Based on the previous results and current authors data, is it possible that Arv1 may function as a scaffold for the initial GPI biosynthetic enzymes including GPI-GnT, GPI-deacetylase, and flippase? The authors could discuss such possibilities.

Reviewer #4: This study describes the composition of the GPI3 protein complex that catalyzes the first step in GPI anchor precursor biosynthesis in Trypanosoma brucei. While this complex has been well described in mammalian and yeast cells, its precise composition in evolutionarily divergent protists, such as T. brucei has not been defined. Using a combination of bioinformatic and label-free proteomic analysis of solubilized, immunoprecipitated T. brucei complex (with TbGPI3-myc as bait) the authors show that the GPI3 complex is differentially stable in different detergents and contains both expected (GPI-3, GPI-5, GPI-19, GPI-2, GPI-1, TbERI1) as well as unanticipated (TbArv-1 and a putative E2-ligase UbCE) proteins. The data also suggest that the complex lacked the non-catalytic DPM2 subunit found in mammalian complexes. The analyses are expertly performed and the finding that the T. brucei complex contains Arv-1 and UbCE homologues has implications for understanding the role of this complex in regulating GPI biosynthesis in crown eukaryotes. Overall, this is a useful study that will be of interest to the parasitology and glycobiology fields.

Minor comments.

Lines 290, 330, TbGPI-19 should be TbGPI-9

6. PLOS authors have the option to publish the peer review history of their article (what does this mean?). If published, this will include your full peer review and any attached files.

Reviewer #1: No

Reviewer #2: No

Reviewer #3: No

Reviewer #4: **Yes: **Malcolm McConville

---

## [Author Response · Author response to Decision Letter 0]

17 Feb 2021

We thank the reviewers for their helpful comments. We respond in to their specific points below:

Reviewer 2 appears satisfied with the paper concluding that “[the paper] will provide a basis for further studies to better understand regulation of GPI biosynthesis in not only in T. brucei but also in human and yeast cells. The manuscript is clearly written and the conclusions are sufficiently supported by experimental evidence.” 

Similarly, Reviewer 4 concludes that “The analyses are expertly performed and the finding that the T. brucei complex contains Arv-1 and UbCE homologues has implications for understanding the role of this complex in regulating GPI biosynthesis in crown eukaryotes. Overall, this is a useful study that will be of interest to the parasitology and glycobiology fields.” 

Other than typographical changes, (such as correcting TbGPI9 to TbGPI19 throughout) no responses appear to be necessary to the reviews of Reviewer 2 and Reviewer 4.

Reviewer #1 says that: The manuscript is a bit thin and borders on the trivial. Response: With all due respect, we would describe the paper as concise and commensurate rather than thin. In responding to the reviewer’s helpful comments (below), we now emphasise just how low-abundance the GPI GnT complex is (see new version of Table 1) and modified text on lines 246-252 & 355-358. We hope this illustrates the non-trivial nature of the optimisations and analyses required, and of the original observations made (viz identifying 6 presumed but unproven complex components and 2 new components in a single study). Anecdotally, two groups (one working on trypanosomes and another on mammalian GPI biosynthesis) have already been in touch to comment on how our discovery of Arv-1 as a GPI GnT component has significantly influenced their research directions. 

Surely the authors could have validated and extended their data to make the paper more generally useful to the field? Some suggestions:

(i) Arv1 is new here - an Arv1 knockdown strain could be generated to test whether the complex as seen on native page is affected in any way. This could also be done with one or other of the subunits - perhaps such strains are available and it would be a matter of expressing myc-tagged Gpi3 in these as a reporter. Response: Unfortunately, there are no mutant strains available for any of the GPI GnT subunits in which to express TbGPI3-Myc to perform these experiments. The lack of orthogonal confirmation that TbArv1 is physically associated with TbGPI3 is acknowledged as a limitation of the study in the Discussion (lines 353-358).

(ii) Is the complex stoichiometric - one copy of each subunit? The native page result (Fig 2) is suggestive, but not definitive. This could be investigated. Response: With respect, determining sub-unit stoichiometry in a complex of such low abundance is beyond the scope of this study. We make no claims about stoichiometry in the paper for this reason. This is acknowledged as a limitation of the study in the Discussion (lines 353-358).

(iii) PIG-A is a single-pass, tail anchored protein, but TbGpi3 appears to have two transmembrane sequences - this could be checked. Response: The reviewer is correct that, according to predictors like TMHMM, TbGPI3 has a putative transmembrane domain close to its N-terminus as well as, like PIG-A, one close to its C-terminus. However, with respect, the purpose of the work in the paper was not to study the topologies of GPI GnT subunits but, as the title suggests, only to identify subunits of the complex. This is acknowledged as a limitation of the study in the Discussion (lines 353-358).

Minor comments:

line 78 - cite Sobering et al. (2004) Cell for additional subunit Eri1. Response: We thank the reviewer for pointing out this omission, the reference is now included (new ref [24]).

Figure 1B, C - the Ponceau-stained membrane shows remarkably few bands for a total lysate, with the dominant band running exactly the same as the myc-tagged protein - this seems odd. Th blot in panel C is of quite poor quality - there are clearer results in Fig 2A. Figure 1 seems unnecessary as the key result in this figure is replicated in Fig 2A. Response: The Ponceau stain (Fig. 1B) illustrates that we cannot usefully load more material than total lysate of 5E6 trypanosomes because distortions are already beginning to show around 50 kDa. This is because SDS-PAGE patterns of total lysates of bloodstream form T. brucei are dominated by disproportionately abundant (glyco)proteins; specifically the highly abundant VSG [1E7 copes per cell, 10% of total cellular protein] that runs just above the 50 kDa marker and tubulin [from the parasite’s densely packed subpellicular microtubule cytoskeleton] that runs just below the 50 kDa marker. Other proteins that have similar apparent molecular weights to these dominant proteins are similarly distorted by them (like TbGPI3-Myc). The absence of any fluorescent signal in Fig. 1C lane 5 (the un-transfected parent cell line) provides evidence of the specificity of the signal. The ‘poorness’ of the Western blot of whole lysate is simply a function of the extremely low abundance of the in-situ tagged (not overexpressed) TbGPI3-Myc product, which is indeed why it was necessary to do a pull-down (from 2E8 cells) to have sufficient signal to work with in the experiments in Fig. 2. Please also see our reply to the next of the reviewer’s points. We have taken the reviewers comments on board and now explain the data better in the revised manuscript (see lines 245-252). We respectfully request we retain the data in (Fig 1) now they are better explained. 

line 251 - it is not necessary to do a pull-down in order to detect complexes in native page - a total lysate could have been run, but perhaps the sample was too dilute? Response: The reviewer is correct, the GPI GnT complex is very low abundance requiring the pull-down/detection approach. An extremely deep total parasite proteome that identified 7148 protein groups – derived from the data in ref [33] and described in the Materials and Methods under ‘Data Analysis’ – allows us to rank each protein from most (rank 1) to least (rank 7148) abundant. Any undetected proteins are ranked 7149. When these total proteome rankings are listed next to the top-8 proteins detected in the TbGPI3-3Myc pull-down (see new version of Table 1) it is, we hope, easier to appreciate how profound the enrichment of all 8 proteins into the pull-downs is. For example, TbArv1 is 7149 (undetectable) in the total proteome but 6th in the (triplicate) TbGPI3-3Myc pull downs. This is now discussed in lines 353-358.

paragraph beginning on line 334 - Arv1 is not essential in yeast cells grown at 25C (Swain, Stukey et al. (2002) JBC; Georgiev, Johansen et al. (2013) Traffic). Although Arv1 may play a role in GPI biosynthesis, perhaps a regulatory role, it is clearly not necessary as otherwise the cells would be inviable. This point should be made clear. Response: These references [39,40] are now included and the reviewer’s helpful point is now made clear in the revised text (lines 360-361).

Reviewer 3 concludes that “The authors identified Arv1 and UbCE proteins as part of GPI-GnT, in addition to TbGPI15, TbGPI9, TbGPI2, TbGPI1, and TbERI1. It is a new finding that two additional factors are componets of GPI-GnT in T. brucei.” but comments “However, the reviewer asks the authors to confirm the mass spectrometric result before publication” because “The major concern is whether Arv1 and UbCE are bound with GPI-GnT directly. The authors only detected two proteins with mass spectrometry. The reviewer suggests to validate the mass spectrometric results. It is possible to detect the interactions between tagged Arv1 or UbCE and TbGPI3-3Myc using western blotting.” Response: We respectfully suggest that dual-tagging and Western blot confirmation of co-precipitation is not necessary. TbArv1 has been enriched in the TbGPI3-Myc3 pull-downs from being undetectable by proteomics (>7148th in the total proteome) to being the 6th most abundant protein in the TbGPI3-Myc3 triplicate pull downs (see also response to Reviewer 1, above). This is now emphasised in the modified version of Table 1 and in the text on lines 356-358. The lack of orthogonal confirmation of TbArv1 – TbGPI3 association is acknowledged as a limitation of the study (lines 353-358).

In lines 340 to 342, the authors wrote that “The complementation of yeast Arv1 mutants by the human Arv1 [38] and recent findings that human Arv1 mutations lead to deficiencies in GPI anchoring [39] [40] strongly suggest a related role in mammalian cells and that it is a component of the mammalian UDP-GlcNAc : PI 1-6 GlcNAc-transferase complex.” It is obvious that Arv1 is involved in the GPI biosynthetic pathway, but cannot specify to be the component of GPI-GnT from the reports. The authors should rewrite the part. Response: We have rewritten this section in-line with the reviewer’s helpful comments – see lines 367-369.

In yeast, it is reported that Arv1 is required for flipping of GPI intermediates or for efficient synthesis of Man1GlcN-acylPI (Okai et al. (2020) FEBS Lett. 594: 2431; Kajiwara et al. (2008) Mol. Biol. Cell 19: 2069). The difference in GPI biosynthetic pathway between yeast and T. brucei is the flipping steps of GPI intermediates. Only the GlcN-PI across the ER lipid bilayer in yeast GPI biosynthesis, whereas several GPI intermediates seem to be flipped/flopped in T. brucei. The authors need to describe the difference of the flipping reaction in yeast and T. brucei GPI biosynthesis, in addition to the difference of inositol-acylation and mannosylation steps. The differences in flipping are now described and referenced [47] (lines 373-376).

Based on the previous results and current authors data, is it possible that Arv1 may function as a scaffold for the initial GPI biosynthetic enzymes including GPI-GnT, GPI-deacetylase, and flippase? The authors could discuss such possibilities. Response:: These possibilities are now discussed (lines 376-378).

In response to the editor: 

1. We have checked Plos One style, including file naming.

2. We have added the strain of T. brucei to line 89.

3. We have discussed the limitations of the study (lines 353-358).

4. We have supplied original uncrossed images (S1_raw_images) in the Supporting Information.

5. Manuscript and submission titles are identical 

6. Table is now integral to the manuscript.

7. Captions are provided for the Supporting Information files.

---

## [Decision Letter · Decision Letter 1]

22 Feb 2021

Proteomic identification of the UDP-GlcNAc : PI α1-6 GlcNAc-transferase subunits of the glycosylphosphatidylinositol biosynthetic pathway of Trypanosoma brucei.

PONE-D-20-39015R1

Dear Dr. Ferguson,

We’re pleased to inform you that your manuscript has been judged scientifically suitable for publication and will be formally accepted for publication once it meets all outstanding technical requirements.

Kind regards,

Ziyin Li, Ph.D.

Academic Editor

PLOS ONE

Additional Editor Comments (optional):

Reviewers' comments:

Reviewer's Responses to Questions

**Comments to the Author**

1. If the authors have adequately addressed your comments raised in a previous round of review and you feel that this manuscript is now acceptable for publication, you may indicate that here to bypass the “Comments to the Author” section, enter your conflict of interest statement in the “Confidential to Editor” section, and submit your "Accept" recommendation.

Reviewer #1: All comments have been addressed

Reviewer #2: All comments have been addressed

Reviewer #3: All comments have been addressed

Reviewer #4: All comments have been addressed

2. Is the manuscript technically sound, and do the data support the conclusions?

Reviewer #1: Yes

Reviewer #2: Yes

Reviewer #3: Yes

Reviewer #4: Yes

3. Has the statistical analysis been performed appropriately and rigorously? 

Reviewer #1: Yes

Reviewer #2: Yes

Reviewer #3: Yes

Reviewer #4: Yes

4. Have the authors made all data underlying the findings in their manuscript fully available?

Reviewer #1: Yes

Reviewer #2: Yes

Reviewer #3: Yes

Reviewer #4: Yes

5. Is the manuscript presented in an intelligible fashion and written in standard English?

Reviewer #1: Yes

Reviewer #2: Yes

Reviewer #3: Yes

Reviewer #4: Yes

6. Review Comments to the Author

Reviewer #1: The authors have responded well to points that I raised and, by adding clarifications to the text and a table, they have made the manuscript much clearer.

Reviewer #2: Line 41, Abstract and line 352, Discussion: E2 enzyme or E2 conjugating-enzyme rather than E2 ligase would be suitable.

I have no other point.

Reviewer #3: The authors addressed the reviewer's concerns. The reviewer does not have any further comments, and now recommends that the manuscript is published in PLOS ONE.

Reviewer #4: I thank the authors for the way in which they have satisfactorily addressed the issues raised by the reviewers. In particular, the additions to the discussion highlight both the limitations and significant implications of the study.

7. PLOS authors have the option to publish the peer review history of their article (what does this mean?). If published, this will include your full peer review and any attached files.

Reviewer #1: No

Reviewer #2: No

Reviewer #3: No

Reviewer #4: No

---

## [Editor Report · Acceptance letter]

11 Mar 2021

PONE-D-20-39015R1 

Proteomic identification of the
UDP-GlcNAc : PI α1-6 GlcNAc-transferase
subunits of the glycosylphosphatidylinositol biosynthetic pathway of *Trypanosoma brucei*. 

Dear Dr. Ferguson:

I'm pleased to inform you that your manuscript has been deemed suitable for publication in PLOS ONE. Congratulations! Your manuscript is now with our production department. 

Kind regards, 

on behalf of

Dr. Ziyin Li 

Academic Editor

PLOS ONE